# Reciprocal interaction between mesenchymal stem cells and transit amplifying cells regulates tissue homeostasis

**Junjun Jing[1,2], Jifan Feng[1], Jingyuan Li[1], Hu Zhao[1], Thach-Vu Ho[1], Jinzhi He[1,2], Yuan Yuan[1], Tingwei Guo[1], Jiahui Du[1], Mark Urata[1], Paul Sharpe[3], Yang Chai[1]***

[1]Center for Craniofacial Molecular Biology, University of Southern California, Los Angeles, United States; [2]State Key Laboratory of Oral Diseases, National Clinical Research Center for Oral Diseases, West China Hospital of Stomatology, Chengdu, China; [3]Department of Craniofacial Development and Stem Cell Biology, Dental Institute, Kings College London, London, United Kingdom

**Abstract** Interaction between adult stem cells and their progeny is critical for tissue homeostasis and regeneration. In multiple organs, mesenchymal stem cells (MSCs) give rise to transit amplifying cells (TACs), which then differentiate into different cell types. However, whether and how MSCs interact with TACs remains unknown. Using the adult mouse incisor as a model, we present *in vivo* evidence that TACs and MSCs have distinct genetic programs and engage in reciprocal signaling cross talk to maintain tissue homeostasis. Specifically, an IGF-WNT signaling cascade is involved in the feedforward from MSCs to TACs. TACs are regulated by tissue-autonomous canonical WNT signaling and can feedback to MSCs and regulate MSC maintenance via Wnt5a/Ror2-mediated non-canonical WNT signaling. Collectively, these findings highlight the importance of coordinated bidirectional signaling interaction between MSCs and TACs in instructing mesenchymal tissue homeostasis, and the mechanisms identified here have important implications for MSC–TAC interaction in other organs.

*For correspondence:
ychai@usc.edu

**Competing interests:** The authors declare that no competing interests exist.

## Introduction

The regulation of tissue homeostasis is a fundamental function of adult stem cells. Once stem cells leave their niche, they commit to a more restricted lineage and/or differentiate into specific cell types. In many tissues and organs, during this process stem cells give rise to transit amplifying cells (TACs), an undifferentiated progenitor population (*Barker et al., 2012*; *Jensen and Watt, 2006*; *Lui et al., 2011*). TACs function as transient but indispensable integrators of stem cell niche components (*Scadden, 2014*; *Xin et al., 2016*; *Rezza et al., 2016*). Several independent studies have revealed that stem cells actively interact with their progeny during tissue homeostasis and regeneration (*Hsu and Fuchs, 2012*; *Hsu et al., 2014a*; *Pardo-Saganta et al., 2015*). For example, regulatory feedback from TACs may instruct stem cells to replenish downstream lineages and serve to coordinate stem cell self-renewal during tissue homeostasis and regeneration of adult ectodermal organs (*Hsu et al., 2014a*). Another study reported that, in the airway epithelium, the parent epithelial stem cells relay Notch signaling to regulate differentiation in their daughter secretory progenitor cells (*Pardo-Saganta et al., 2015*). To date, however, these pioneering studies have exclusively focused on ectodermal organs, and it remains unknown how mesenchymal stem cells (MSCs) interact with TACs to maintain tissue homeostasis.

The mouse incisor provides an ideal model for studying the interaction of MSCs and TACs because the stem cells and their niche are retained throughout the mouse's life to support continuous incisor growth, and the anatomical locations of the dental pulp stem cells and TAC populations have recently been clearly defined (*An et al., 2018*; *Sharpe, 2016*; *Zhao et al., 2014*; *Kaukua et al., 2014*). The mouse incisor is comprised of an outer layer of enamel with dentin underneath and an inner chamber of dental pulp that contains vasculature and nervous tissue. The epithelial and mesenchymal compartments of the incisor both replenish all of their cells within the course of a month (*Zhao et al., 2014*). The continuous turnover of mesenchymal tissue is supported by MSCs in the proximal region of the mouse incisor. Our previous study demonstrated that quiescent Gli1+ dental pulp cells are typical MSCs in the mouse incisor and they generate TACs that can be found in the immediately adjacent region (*Zhao et al., 2014*). Gli1+ MSCs can continuously populate the incisor mesenchyme throughout the animal's lifetime (*Zhao et al., 2014*). TACs actively proliferate, giving rise to committed preodontoblasts, terminally differentiated odontoblasts, and dental pulp cells. Recent studies have also identified that polycomb repressive complex 1 (PRC1) regulates the TACs via WNT/β-catenin signaling, and that Axin2 expression identifies the TAC population, which is unable to maintain itself through self-renewal (*An et al., 2018*).

In this study, we took advantage of specific molecular markers that allowed us to identify and target either MSCs or TACs in the adult mouse incisor *in vivo* and we found that MSCs and TACs have distinct genetic programs that may help them define their *in vivo* identities and interact with each other. We learned that MSCs feedforward to TACs through an IGF-WNT signaling cascade and the control of TAC fate depends on tissue-autonomous canonical WNT signaling. In parallel, TACs produce Wnt5a, which provides feedback to MSCs via Ror2-mediated non-canonical WNT signaling. Our study provides *in vivo* evidence of the reciprocal interaction between MSCs and TACs in mesenchymal tissue homeostasis and highlights the molecular regulatory mechanism that governs this interaction. The mechanisms identified in this study could potentially apply to other organs, such as long bone, where MSC and TAC interaction is not well understood but may also be crucial for maintaining tissue homeostasis and regeneration.

## Results

### Anatomical and molecular identities of MSCs and TACs in the mouse incisor

In order to study the interaction between MSCs and TACs, we first confirmed their *in vivo* locations using recently published markers. In addition to Axin2, TACs are also identifiable by their active proliferation status (*Zhao et al., 2014*), consistent with our findings that Ki67+ and Axin2+ populations both reside in the TAC region (*Figure 1A and B*). Therefore, we used both Ki67 and Axin2 as TAC markers. Gli1 is a known dental MSC marker in the adult mouse incisor (*Zhao et al., 2014*). Our data clearly indicate that Gli1+ MSCs and TACs are mutually exclusive cell populations in adult incisors (*Figure 1C and D*), consistent with our previous findings (*Zhao et al., 2014*). This allowed us to use *Gli1-CreER^{T2}* and *Axin2-CreER^{T2}* to target MSCs and TACs in the mouse incisor, respectively. The close proximity between the MSCs and TACs in the mouse incisor suggests a physical environment conducive of cell–cell interaction between these two populations (*Figure 1E*).

### IGF ligand and binding proteins are highly enriched in MSCs

Identifying genetic signatures of MSCs and TACs is critical to understanding the genetic programs involved in establishing and maintaining their identities as well as uncovering how these two populations interact. To identify genetic profiles of MSCs and TACs, we used laser capture microdissection (LCM) to collect the MSC region from *Gli1-CreER^{T2};Rosa26^{<fs-tdTomato>}* mice and the TAC region from *Axin2-CreER^{T2};Rosa26^{<fs-tdTomato>}* mice shortly after labeling, followed by unbiased RNA sequencing analysis of these samples. We found that specific signaling pathways were preferentially enriched in either TACs or MSCs based on Ingenuity Pathway Analysis (IPA, QIAGEN). We first analyzed the signaling enriched in TACs. As expected for Axin2+ TACs, WNT/β-catenin signaling was one of the top enriched pathways (*Figure 2A*). Cell cycle regulation signaling was also highly active, consistent with the high proliferative activity of TACs. Moreover, mTOR and EIF2 signaling, which are both important for cell proliferation, were also enriched in TACs (*Figure 2A*). We focused

on IGF signaling based on previous reports that it plays a role in stem cell homeostasis (*Youssef et al., 2017*; *Ziegler et al., 2015*). We investigated the ligands and receptors of IGF signaling that were found among the significantly differentially expressed genes between incisor MSCs and TACs identified by RNA sequencing. Although IGF signaling was highly active in TACs, we found that IGF signaling molecules including IGF binding protein (*Igfbp3*) and the IGF ligand, insulin-like growth factor 2 (*Igf2*), were all highly enriched in MSCs (*Figure 2B*). RNAscope *in situ* hybridization (ISH) confirmed that *Igfbp3* was highly expressed in MSCs outside of the TAC region and dental follicle *in vivo* (*Figure 2C and D*), and *Igf2* also appeared to be highly expressed in a similar pattern (*Figure 2E and F*). Interestingly, consistent with the pathway analysis (*Figure 2A*), expression of *Igf1r*, an IGF receptor, was detectable in the mouse incisor mesenchyme overlapping the TAC region (*Figure 2G and H*), indicating that MSC- and dental follicle-derived IGF signals may affect TAC fate through the IGF signaling pathway.

## Inactivation of IGF signaling leads to diminished TACs and compromised WNT signaling in the incisor mesenchyme

In order to investigate the potential role of IGF signaling in MSC–TAC interaction, we generated *Axin2-CreER^{T2}*;*Igf1r^{fl/fl}* mice, in which *Igf1r* could be deleted in Axin2+ TACs via tamoxifen induction. We found that the number of TACs in the mesenchyme was diminished 3 weeks after induction (*Figure 3C–E*), indicating that IGF signaling is critical for TAC proliferation. In addition, the dentin became thicker in *Axin2-CreER^{T2}*;*Igf1r^{fl/fl}* mice (*Figure 3A and B*), indicating premature differentiation of TACs, suggesting that IGF signaling pathway may regulate TAC fate through balancing proliferation and differentiation. We found that the number of TACs in the incisor epithelium of *Axin2-CreER^{T2}*;*Igf1r^{fl/fl}* mice was also reduced 3 weeks after induction (*Figure 3C and D*), suggesting epithelial–mesenchymal interaction occurred at this stage. Because *Axin2-CreER^{T2}* is highly active in the TAC region of the incisor mesenchyme but not the epithelium, we hypothesized that the effect from the mesenchyme is primary. In order to test whether loss of *Igf1r* in Axin2+ TACs in the incisor mesenchyme was indeed the primary cause of the TAC loss, we analyzed the change of TACs in *Axin2-CreER^{T2}*;*Igf1r^{fl/fl}* incisors at an earlier time point and found that the number of TACs in the mesenchyme was already reduced whereas differentiation (indicated by *Dspp*, which is a marker for odontoblasts) was enhanced 1 week after induction (*Figure 3—figure supplement 1*). However, the number and differentiation status of TACs in the incisor epithelium (indicated by *Amelx*, which is a marker for ameloblasts) remained unchanged in *Axin2-CreER^{T2}*;*Igf1r^{fl/fl}* mice at this time point (*Figure 3—figure supplement 1*), suggesting that epithelial–mesenchymal interaction occurs later and the effect from the incisor mesenchyme is primary. Moreover, we found that *Axin2* expression was reduced in TACs, suggesting that canonical WNT signaling is likely to be downstream of IGF signaling in the regulation of TACs (*Figure 3F and G*). Consistent with reduced *Axin2* expression, we found that other WNT downstream genes such as c-Myc and active β-catenin were also downregulated

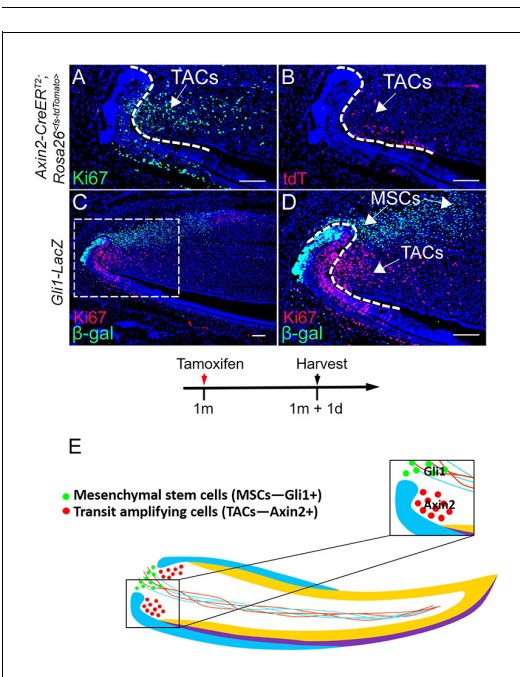

**Figure 1.** *In vivo* anatomical and molecular identities of mesenchymal stem cells (MSCs) and transit amplifying cells (TACs) in the mouse incisor. (**A and B**) Ki67 staining (green), tdTomato visualization (red), and DAPI staining (blue) of incisors from 1-month-old *Axin2-CreER^{T2}*;*Rosa26^{<fs-tdTomato>}* mice 1 day after tamoxifen (TM) induction. (**C and D**) β-gal staining (green) and Ki67 immunofluorescence (red) of incisors from 1-month-old *Gli1-LacZ* mice. Box in (**C**) is shown enlarged in (**D**). (**E**) Schematic diagram of MSCs and TACs in the mouse incisor. Arrows indicate positive signal. Induction protocol schematic indicates tamoxifen (TM) administration and sample collection time. The white dashed lines outline the cervical loop. Scale bars, 100 μm.

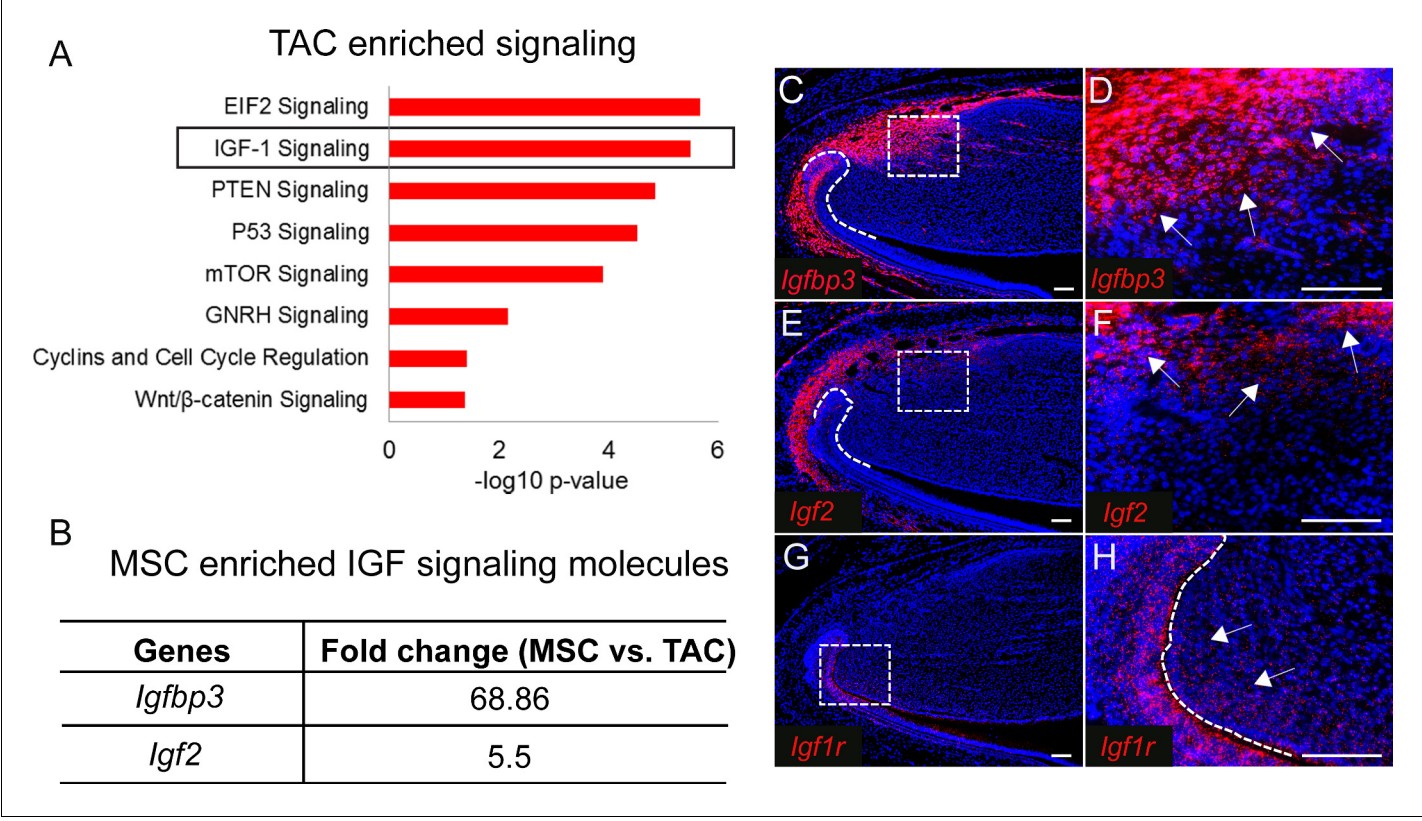

**Figure 2.** IGF ligand and binding proteins are highly enriched in mesenchymal stem cells (MSCs). (**A**) Top eight signaling pathways enriched in transit amplifying cells (TACs) identified by Ingenuity Pathway Analysis. (**B**) IGF signaling molecules enriched in MSCs. (**C–H**) RNAscope (red) of *Igfbp3* (**C and D**), *Igf2* (**E and F**), and *Igf1r* (**G and H**) in incisors of 1-month-old control mice. Boxes in (**C, E, and G**) shown magnified in (**D, F, and H**), respectively. Arrows indicate positive signal. The white dashed lines outline the cervical loop. Scale bars, 100 μm.

(*Figure 3—figure supplement 2*). Therefore, we concluded that MSCs and the dental follicle secrete Igf2 ligand to regulate TACs through an IGF-WNT signaling cascade (*Figure 3H*). We found that the phenotype in *Axin2-CreER^{T2};Igf1r^{fl/fl}* incisors persisted 5 months after induction, suggesting disturbed tissue homeostasis of the incisor at later stages (*Figure 3—figure supplement 3*).

## WNT signaling molecules are enriched in TACs but not MSCs

Previous studies have shown that WNT signaling is important for TAC regulation (*An et al., 2018*). Due to the close anatomical proximity of MSCs and TACs, we further analyzed whether WNT ligands are secreted by MSCs in a paracrine manner or by TACs in an autocrine manner. Surprisingly, TACs, not MSCs, seem to be the main source of both canonical and non-canonical WNT ligands (*Wnt10a* and *Wnt5a*, respectively) (*Figure 4A*). *Wnt10a* was detectable in the TAC region in the mesenchyme, although it was also expressed in odontoblasts and the epithelium (*Figure 4B and C*). *Wnt5a* was also highly expressed in TACs as well as in odontoblasts and in the dental mesenchyme, but excluded from the MSC region in the incisor (*Figure 4D and E*). To test whether any other WNT ligands were expressed in MSCs or TACs, we assessed the expression patterns of the 17 other WNT ligands. None were enriched in either MSCs or TACs: some were only expressed in the epithelium but not in the mesenchyme, such as *Wnt3a* and *Wnt4*, whereas others were undetectable in TACs in both the epithelium and the mesenchyme (*Figure 4—figure supplement 1*). This result suggests that both canonical and non-canonical WNTs in the incisor mesenchyme are mainly TAC-derived. To search for the cells that are responsive to the WNTs, we analyzed enriched pathways in

both TACs and MSCs. Interestingly, canonical WNT/β-catenin signaling was enriched in TACs (*Figure 2A*) whereas non-canonical NFAT signaling was enriched in MSCs (*Figure 4F*).

## Loss of WNT signaling in odontoblasts has no effect on MSCs or TACs

To test whether WNT signaling in odontoblasts has an effect on MSCs or TACs, we generated *Dmp1-Cre;Wls$^{fl/fl}$;Gli1-LacZ* mice in which odontoblasts are unable to secrete the WNT ligand. We found that there was no significant difference between *Gli1-LacZ* and *Dmp1-Cre;Wls$^{fl/fl}$;Gli1-LacZ*

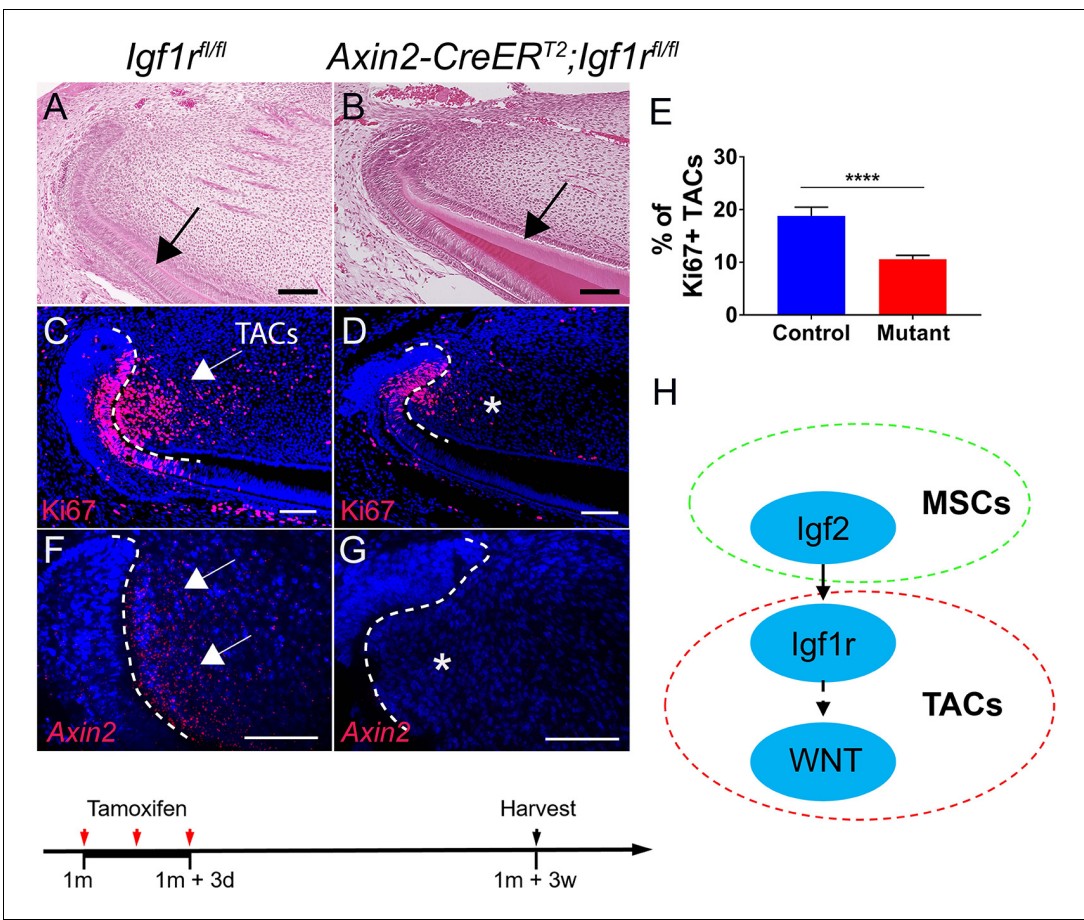

**Figure 3.** Inactivation of IGF signaling leads to transit amplifying cell (TAC) loss in the incisor mesenchyme. (A–D) H and E staining and Ki67 immunofluorescence (red) of incisors from *Igf1r$^{fl/fl}$* (control) and *Axin2-CreER$^{T2}$;Igf1r$^{fl/fl}$* mice. Black arrow in (A) indicates normal dentin and black arrow in (B) indicates thicker dentin. White arrow in (C) indicates a positive signal and asterisk in (D) indicates diminished signal. (E) Quantitation of the percentage of Ki67+ TACs from (C and D). (F and G) RNAscope staining of *Axin2* expression (red) in incisors from *Igf1r$^{fl/fl}$* (control) and *Axin2-CreER$^{T2}$;Igf1r$^{fl/fl}$* mice. White arrow in (F) indicates a positive signal and asterisk in (G) indicates absence of a signal. (H) Diagram depicts our model of the Igf2-WNT signaling cascade. Quantitative data are presented as mean ± SD. ****, p<0.0001. Schematic at the bottom indicates induction protocol. The white dashed lines outline the cervical loop. Four mice with four sections within each mouse per group were used to quantify Ki67+ cells. Scale bars, 100 μm.

The online version of this article includes the following source data and figure supplement(s) for figure 3:

**Source data 1.** Source data for *Figure 3E*.

**Figure supplement 1.** Effect of the incisor mesenchyme is primary in transit amplifying cell (TAC) loss in *Axin2-CreER$^{T2}$;Igf1r$^{fl/fl}$* incisors.

**Figure supplement 1—source data 1.** Source data for *Figure 3—figure supplement 1C and D*.

**Figure supplement 2.** Canonical WNT signaling is downregulated after loss of Igf1r in the transit amplifying cells (TACs) of the mouse incisor.

**Figure supplement 3.** Long-term phenotype of *Axin2-CreER$^{T2}$;Igf1r$^{fl/fl}$* incisors.

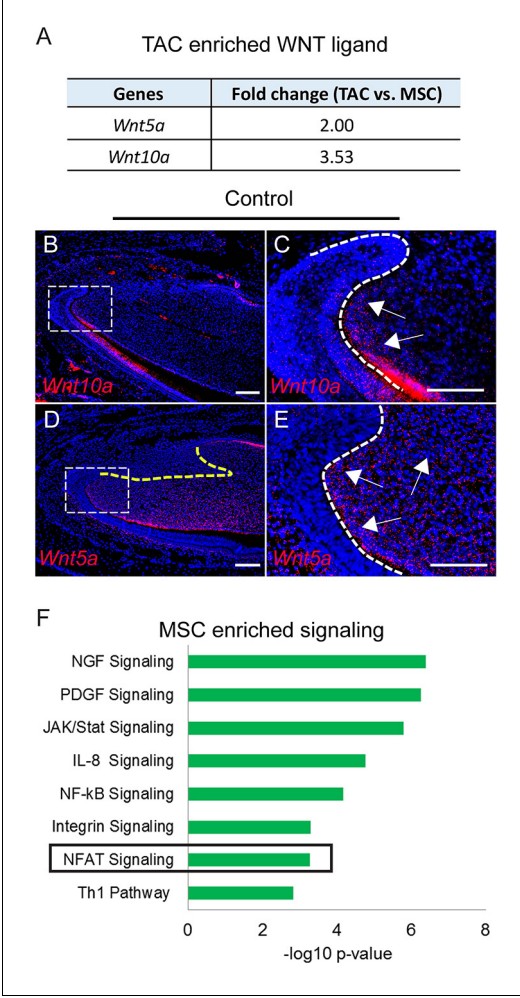

**Figure 4.** Signaling enriched in mesenchymal stem cells (MSCs) and WNT ligand enriched in transit amplifying cells (TACs). (**A**) *Wnt10a* and *Wnt5a* are enriched in TACs. (**B–E**) RNAscope (red) of *Wnt10a* and *Wnt5a* in incisors from 1-month-old control mice. (**F**) Top eight signaling pathways enriched in MSCs identified by Ingenuity Pathway Analysis. Boxes in (**B** and **D**) are shown magnified in (**C** and **E**), respectively. Arrows indicate positive signal. The white dashed lines outline the cervical loop. The yellow dashed line outlines the MSC region. Scale bars, 100 µm.

The online version of this article includes the following source data and figure supplement(s) for figure 4:

**Figure supplement 1.** WNT ligand expression in adult incisors.

**Figure supplement 2.** Loss of WNT signaling in odontoblasts has no effect on mesenchymal stem cells (MSCs) or transit amplifying cells (TACs).

**Figure supplement 2—source data 1.** Source data for *Figure 4—figure supplement 2E and J*.

**Figure supplement 3.** Loss of *Wls* in transit amplifying cells (TACs) results in diminished TACs and mesenchymal stem cells (MSCs) in the incisors of *Axin2-CreER^T2^;Wls^fl/fl^;Gli1-LacZ* mice.

*Figure 4 continued on next page*

mice in the number of TACs and MSCs, suggesting that loss of WNT signaling in odontoblasts has no effect on MSCs or TACs (*Figure 4—figure supplement 2*).

## Axin2+ TACs are regulated via tissue-autonomous canonical WNT signaling

In order to investigate the effect of the WNT ligand in Axin2+ TACs, we generated *Axin2-CreER^T2^;Wls^fl/fl^;Gli1-LacZ* mice, in which Axin2+ cells are not able to secrete the WNT ligand. We confirmed that Wls was expressed in the TAC region of control mice, whereas it was undetectable in the TAC region of *Axin2-CreER^T2^;Wls^fl/fl^;Gli1-LacZ* mice (*Figure 4—figure supplement 3A and B*). Interestingly, we found that the *Axin2-CreER^T2^;Wls^fl/fl^;Gli1-LacZ* mouse incisors were significantly shorter 1 month after tamoxifen induction (*Figure 4—figure supplement 3C and D*), suggesting that loss of *Wls* affected their tissue turnover and homeostasis. More importantly, loss of *Wls* in Axin2+ cells resulted in a loss of TACs, as assessed by Ki67 staining (*Figure 4—figure supplement 3E and F*). In order to confirm that TACs were indeed diminished, we also examined the expression of *Fgf10*, which is secreted by TACs in the mesenchyme (*Kuang-Hsien Hu et al., 2014*; *Wang et al., 2007*), and found that *Fgf10* was greatly reduced in *Axin2-CreER^T2^;Wls^fl/fl^;Gli1-LacZ* mice (*Figure 4—figure supplement 3G and H*). In addition, in *Axin2-CreER^T2^;Wls^fl/fl^;Gli1-LacZ* mice, there was a reduction of Gli1+ MSCs (*Figure 4—figure supplement 3I–L*), indicating that TAC-derived WNT signaling may play a role in MSC maintenance. Therefore, TACs may secrete WNTs to regulate the fates of TACs and MSCs.

Axin2 is a readout of canonical WNT signaling. Consistent with the expression of *Axin2* in TACs, we found that *Lrp5*, a canonical WNT receptor, was highly expressed in TACs (*Figure 4—figure supplement 4A and B*), whereas *Fzd2*, *Fzd4*, and *Fzd6* were almost undetectable in the incisor mesenchyme (*Figure 4—figure supplement 4C–H*). These data suggest that Axin2+ TACs are regulated via tissue-autonomous canonical WNT signaling.

## TACs feedback to MSCs via Wnt5a/Ror2-mediated non-canonical WNT signaling

Because Gli1+ MSCs were reduced in *Axin2-CreER^T2^;Wls^fl/fl^;Gli1-LacZ* mice in which Axin2+

*Figure 4 continued*

**Figure supplement 4.** WNT receptor expression in adult incisors.

TACs are unable to secrete WNT ligands, we hypothesized that TAC-derived WNTs can act on MSCs via a paracrine mechanism. Interestingly, delayed molar development has been observed in *Wnt5a*$^{-/-}$ mutant mice, consistent with the notion that Wnt5a-mediated non-canonical WNT signaling plays a role during tooth development (*Lin et al., 2011*), indicating that Wnt5a may play an important role in feedback from TACs to MSCs in the maintenance of incisor tissue homeostasis. To test our hypotheses, we generated *Axin2-CreER*$^{T2}$*;Wnt5a*$^{fl/fl}$*;Gli1-LacZ* mice. We first examined the efficiency of *Wnt5a* deletion in *Axin2-CreER*$^{T2}$*;Wnt5a*$^{fl/fl}$ incisors and confirmed that *Wnt5a* was efficiently deleted in the TAC region of the mutant incisors 3 days after tamoxifen induction (*Figure 5—figure supplement 1*). We found that TACs were not affected in *Axin2-CreER*$^{T2}$*;Wnt5a*$^{fl/fl}$*;Gli1-LacZ* mice 3 weeks after tamoxifen induction (*Figure 5A–C*) but by that time the number of Gli1+ MSCs was already reduced in these *Wnt5a* mutants (*Figure 5D–H*), suggesting that Wnt5a acts as a paracrine WNT ligand to regulate the maintenance of Gli1+ MSCs. We also used label retaining cells (LRCs), which serve as a hallmark of stem cells, to confirm the decrease of MSCs in the incisor of *Wnt5a* mutant mice. We found that loss of *Wnt5a* resulted in diminished LRCs in the MSC region in *Axin2-CreER*$^{T2}$*;Wnt5a*$^{fl/fl}$ mice (*Figure 5—figure supplement 2*). Thicker dentin and diminished TACs were observed in *Axin2-CreER*$^{T2}$*;Wnt5a*$^{fl/fl}$*;Gli1-LacZ* mice at later stages (*Figure 5—figure supplement 3*), indicating disturbed homeostasis of the incisor after the disruption of TACs feedback to MSCs. To investigate the signaling mechanism of TAC feedback to MSCs, we examined the expression of *Ror2*, which serves as a key receptor for Wnt5a. We found that *Ror2* was highly expressed in the MSC region of the incisor mesenchyme (*Figure 6A*), consistent with regulation of Gli1+ MSCs by Wnt5a-mediated non-canonical WNT signaling feedback through Ror2. In order to investigate the function of Ror2-mediated non-canonical WNT signaling in the feedback from TACs to MSCs, we generated *Gli1-CreER*$^{T2}$*;Ror2*$^{fl/fl}$*;Gli1-LacZ* mice, in which *Ror2* was successfully deleted in the Gli1+ cells (*Figure 6B*). Three weeks after tamoxifen induction, Gli1+ MSCs were reduced in the incisor mesenchyme of *Gli1-CreER*$^{T2}$*;Ror2*$^{fl/fl}$*;Gli1-LacZ* mice, indicating that Ror2-mediated non-canonical WNT signaling is critical for the maintenance of Gli1+ MSCs (*Figure 6C–G*). The reduction of MSCs was further confirmed by LRCs (*Figure 6—figure supplement 1*). Taken together, our data indicate that non-canonical WNT signaling mediated by Wnt5a and Ror2 is involved in the TAC feedback to MSCs in regulating tissue homeostasis.

## Discussion

The homeostatic maintenance of self-renewing tissue relies not only on the function of resident somatic stem and progenitor cells, but also on the interaction between these cell populations. In the present work, we studied the adult mouse incisor to uncover novel insights into the interaction between MSCs and TACs (*Figure 7*). Our data demonstrate that the IGF-WNT signaling cascade is crucial for MSC feedforward to TACs. In parallel, TACs feedback to MSCs via Wnt5a/Ror2-mediated non-canonical WNT signaling. Distinct genetic programs within TACs and MSCs may determine their *in vivo* identities and facilitate this dynamic reciprocal interaction to maintain mesenchymal tissue homeostasis.

### Molecular signaling between stem cells and TACs

The question of which genetic programs determine the identities of stem cells and their progeny remains open. Previous studies have addressed this issue in epithelial stem cells. Hsu and colleagues found that TAC-derived SHH is critical in the hair follicle epithelium for stem cell activation and proliferation (*Hsu et al., 2014b*). Another recent study defined gene signatures specific to stem cells and TACs in the hair follicle using RNA sequencing and subsequent transcriptomic analysis (*Rezza et al., 2016*). Single-cell RNA sequencing and lineage-tracing studies have revealed that hair follicle TACs represent a heterogeneous population of progenitors with distinct molecular signatures, reflecting different local signals and intercellular interactions according to their spatial locations (*Yang et al., 2017*). Hair follicle TACs appear to be lineage-biased, as different cells preferentially give rise to seven cell types. In the dental epithelium, Hippo pathway components Yap

and Taz are expressed in TACs and prevent their premature differentiation; Yap/Taz activate mTOR signaling to promote TAC proliferation in response to integrin/FAK signaling (*Hu et al., 2017*).

Despite this evolving understanding of epithelial stem cells and TACs, the distinct genetic signatures distinguishing MSCs from TACs are largely unknown. In our study, we identified genes that are differentially expressed in MSCs and TACs through RNA sequencing analysis. We found that several

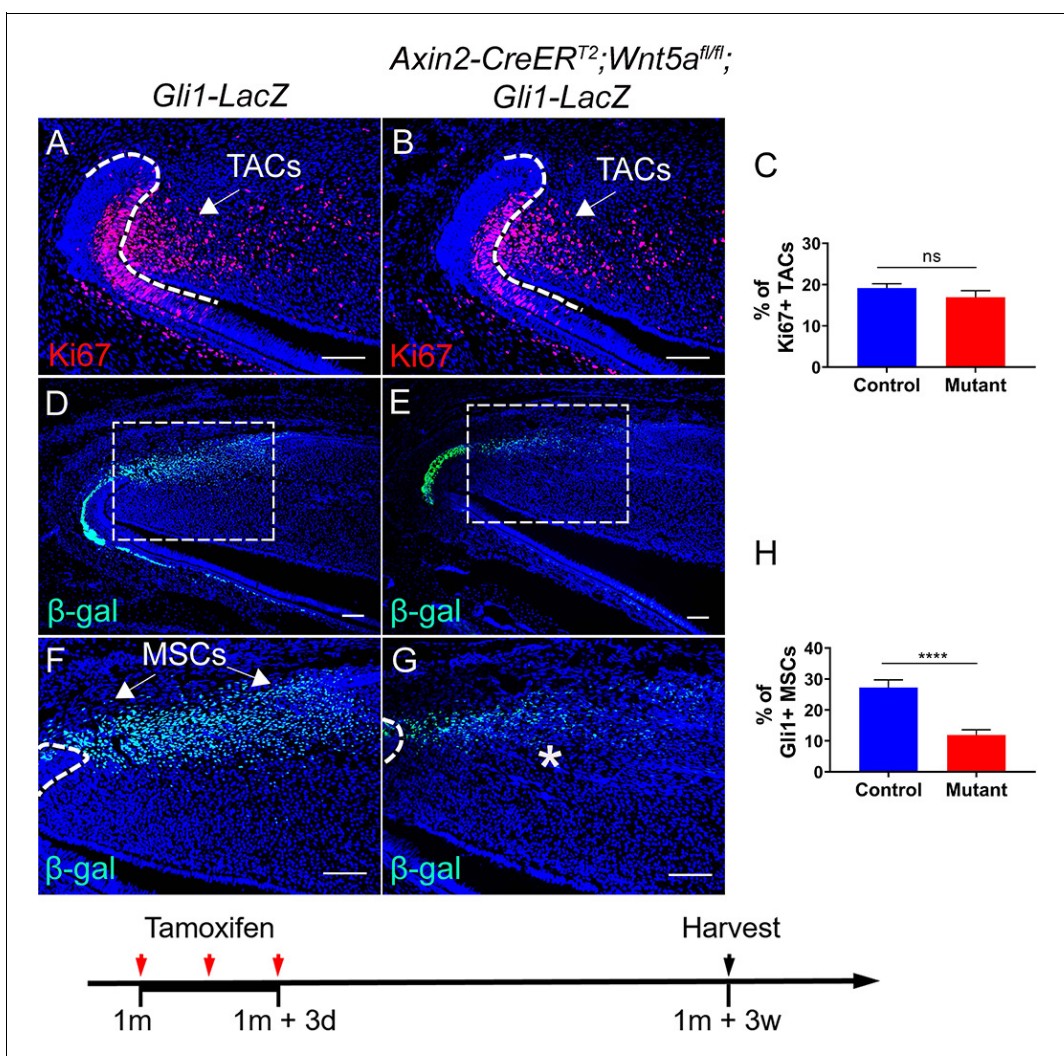

**Figure 5.** Loss of Wnt5a results in loss of Gli1+ mesenchymal stem cells (MSCs). (**A and B**) Ki67 staining (red) of incisors from *Gli1-LacZ* (control) and *Axin2-CreER^{T2};Wnt5a^{fl/fl};Gli1-LacZ* mice 3 weeks after tamoxifen induction. (**C**) Quantitation of the percentage of Ki67+ transit amplifying cells (TACs) from (**A and B**). (**D–G**) β-gal staining (green) of incisors from 1-month-old *Gli1-LacZ* and *Axin2-CreER^{T2};Wnt5a^{fl/fl};Gli1-LacZ* mice 3 weeks after tamoxifen induction. Boxes in (**D and E**) are shown magnified in (**F and G**), respectively. (**H**) Quantitation of the percentage of Gli1+ cells per higher magnification section (**F and G**) of *Gli1-LacZ* and *Axin2-CreER^{T2};Wnt5a^{fl/fl};Gli1-LacZ* incisor mesenchyme. Schematic at the bottom indicates induction protocol. The white dashed lines outline the cervical loop. Arrows indicate positive signal and asterisks indicate diminished signal. All quantitative data are presented as mean ± SD. ns, no significance. ****, $p<0.0001$. Four mice with four sections within each mouse per group were used to quantify Ki67+ cells. Gli1+ cells in the proximal region between the two cervical loops were counted in the mouse incisor. Scale bars, 100 μm.

The online version of this article includes the following source data and figure supplement(s) for figure 5:

**Source data 1.** Source data for *Figure 5C and H*.
**Figure supplement 1.** Validation of *Wnt5a* knockout efficiency in *Axin2-CreER^{T2};Wnt5a^{fl/fl}* incisors.
**Figure supplement 2.** Loss of Wnt5a results in diminished EdU+ label retaining cells (LRCs).
**Figure supplement 2—source data 1.** Source data for *Figure 5—figure supplement 2E*.
**Figure supplement 3.** Loss of Wnt5a results in thicker dentin and diminished transit amplifying cells (TACs).

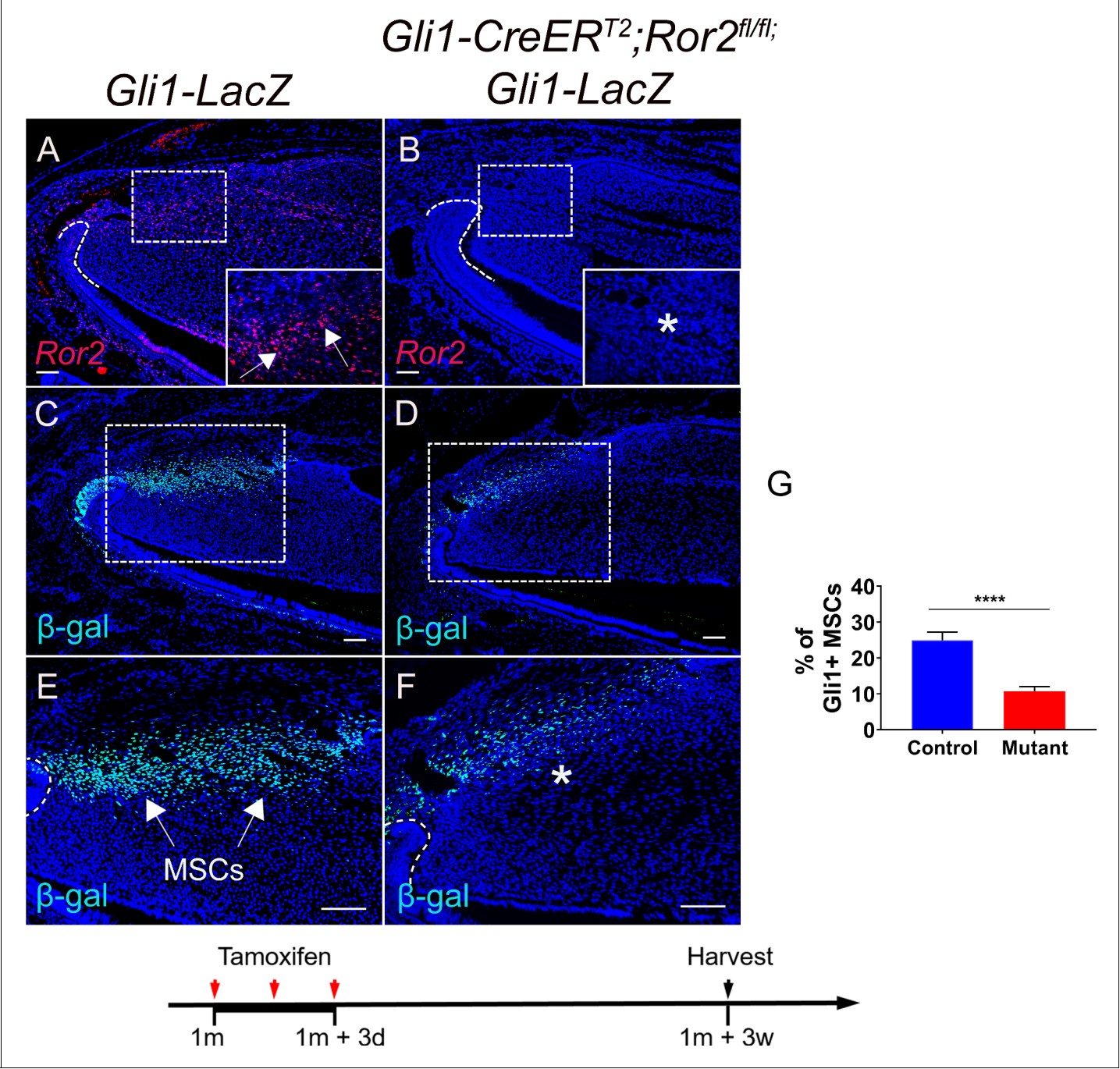

**Figure 6.** Ror2-mediated non-canonical Wnt signaling regulates mesenchymal stem cell (MSC) maintenance in the incisor. (**A and B**) RNAscope (red) of *Ror2* in incisors from *Gli1-LacZ* (control) and *Gli1-CreER^{T2};Ror2^{fl/fl};Gli1-LacZ* 1-month-old mice. Insets in **A** and **B** show magnified images of the MSC region. (**C–F**) β-gal staining (green) of incisors from *Gli1-LacZ* and *Gli1-CreER^{T2};Ror2^{fl/fl};Gli1-LacZ* mice. Boxes in **C** and **D** are shown magnified in **E** and **F**, respectively. (**G**) Quantification of the percentage of Gli1+ cells per higher magnification section of *Gli1-LacZ* (control) and *Gli1-CreER^{T2};Ror2^{fl/fl};Gli1-LacZ* (mutant) mouse incisor mesenchyme in **E** and **F**. Gli1+ cells in the proximal region between the two cervical loops were counted in the mouse incisor. Quantitative data are presented as mean ± SD. ****, p<0.0001. Arrows indicate positive signal and asterisks indicate diminished signal. Schematic at the bottom indicates induction protocol. Scale bars, 100 μm.

The online version of this article includes the following source data and figure supplement(s) for figure 6:

**Source data 1.** Source data for *Figure 6G*.
**Figure supplement 1.** Loss of Ror2 results in diminished EdU+ label retaining cells (LRCs).
**Figure supplement 1—source data 1.** Source data for *Figure 6—figure supplement 1E*.

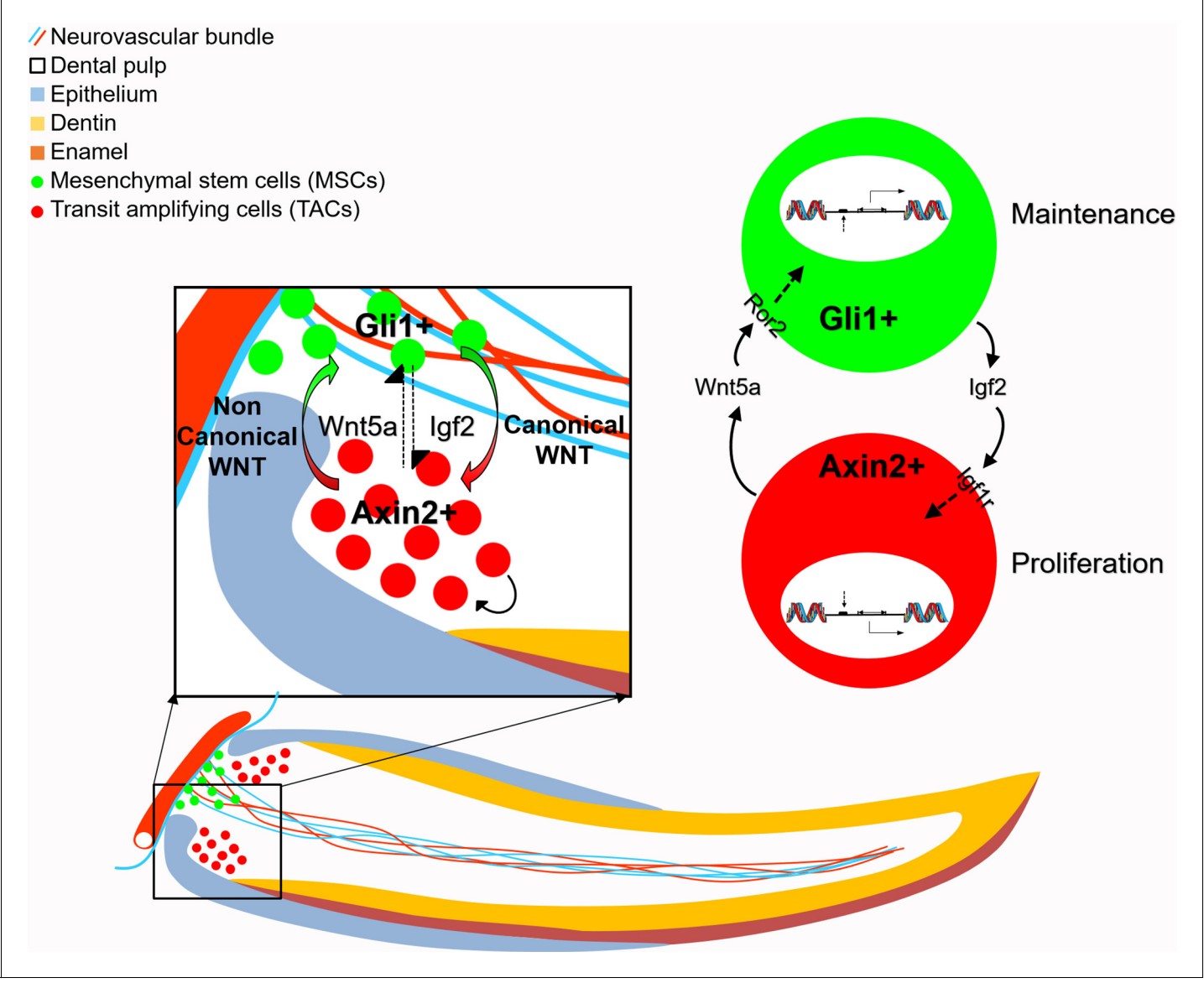

**Figure 7.** Schematic diagram of bidirectional interaction between transit amplifying cells (TACs) and mesenchymal stem cells (MSCs). Igf2 secreted from Gli1+ MSCs binds to Igf1r in the Axin2+ TACs, activating target gene expression and regulate TAC proliferation. Wnt5a serves as a non-canonical WNT ligand, feeding back to MSCs through Ror2 to activate downstream gene expression to regulate the maintenance of MSCs. Thus, MSCs and TACs dynamically interact with each other to maintain mesenchymal tissue homeostasis.

pathways related to cell proliferation, such as the mTOR signaling, were highly enriched in TACs (*Laplante and Sabatini, 2012*; *Saxton and Sabatini, 2017*), consistent with their high proliferative activity. On the other hand, growth factor signaling pathways were enriched in MSCs, including IGF, which has been previously implicated in stem cell homeostasis (*Youssef et al., 2017*; *Ziegler et al., 2015*; *Xian et al., 2012*), suggesting that MSCs may secrete these factors to orchestrate their niche environment. Finally, these different signaling pathways may work together to build the identities of MSCs and TACs and facilitate the interaction between them to maintain tissue homeostasis. This model may serve as a stepping stone in our quest to gain a better understanding of MSCs and their interaction with neighboring cells in regulating tissue homeostasis.

## Stem cell–TAC interaction in tissue homeostasis and regeneration

TACs are important progenitors that represent an intermediate step between MSCs and their differentiated progeny. Several recent findings have indicated that TACs are crucial to tissue regeneration. Specifically, TACs coordinate tissue production, govern stem cell behaviors, and instruct niche remodeling (*Zhang et al., 2016*; *Zhang and Hsu, 2017*; *Carrieri et al., 2017*). Recent studies also suggest that TACs may maintain homeostasis long-term in some cases, and thus may not actually be transient in nature (*Busch et al., 2015*; *Sun et al., 2014*). These findings highlight several previously unrecognized functions of TACs, as well as the need to reevaluate their role in different biological states including homeostasis, regeneration, injury repair, and disease.

Recently, it has been suggested that some differentiated progeny of stem cells may serve as niche components and that interaction between stem cells and their progeny is critical for tissue homeostasis and regeneration. In the airway epithelium, stem/progenitor cells pass on a signal to their progeny that is necessary for daughter cell maintenance (*Pardo-Saganta et al., 2015*). Moreover, some differentiated progeny of stem cells can also provide feedback to regulate their stem cell parents. In the intestine, terminally differentiated Paneth cells localized between crypt stem cells promote stem cell self-renewal (*Sato et al., 2011*). Megakaryocytes regulate hematopoietic stem cell maintenance by regulating quiescence (*Bruns et al., 2014*). The progeny of stem cells might also be able to replenish them to maintain tissue homeostasis. For example, differentiated airway epithelial progenitor cells have been shown to revert into stem cells *in vivo* after the airway stem cells are ablated (*Tata et al., 2013*). However, these studies have investigated only epithelial tissues and there is a lack of information about whether mesenchymal cells may participate in similar processes. In this study, we found that TACs in the mesenchyme have a profound impact on MSCs *in vivo* and that MSCs in turn instruct TACs via feedforward signals, indicating the existence of common regulatory mechanisms in the MSC niche environment.

## Canonical and non-canonical WNT signaling work together to maintain MSCs and control TAC fate

Non-canonical WNT pathways are not well understood compared to their canonical counterpart, and most findings center around their ability to disrupt canonical WNT/β-catenin signaling (*Lien and Fuchs, 2014*). During molar tooth development, Wnt5a plays a key role in regulating growth, patterning, and development (*Lin et al., 2011*). Previous study has also shown that non-canonical WNT signaling is required to maintain adult stem cells, including hematopoietic stem cells (*Sugimura et al., 2012*), and to mediate cell–cell interactions in some organs. Non-canonical WNT signaling might also regulate the interaction between MSCs and hematopoietic stem cells (*Sugimura and Li, 2010*). Wnt5a-Ror2 signaling between cells in the osteoblast lineage and precursors of osteoclasts has been shown to enhance osteoclastogenesis and maintain homeostasis during adult bone remodeling (*Maeda et al., 2012*). Depending on the receptor context, Wnt5a can either activate or inhibit β-catenin signaling (*Mikels and Nusse, 2006*; *van Amerongen et al., 2012*). Cdc42-mediated non-canonical WNT signaling has been shown to regulate neural stem cell quiescence in homeostasis and after demyelination (*Chavali et al., 2018*). NFATc1 is involved in the balance of quiescence and proliferation in skin stem cells (*Goldstein et al., 2014*; *Horsley et al., 2008*), whereas NFATc4 regulates neural stem cells (*Moreno et al., 2015*).

Here we report that Wnt5a acts through Ror2 to facilitate the feedback from TACs to MSCs in the incisor mesenchyme, indicating a role for non-canonical WNT signaling in MSCs. TACs also show strong canonical WNT signaling activity. The differential expression of WNT receptors/intracellular mediators in the adult incisor mesenchyme may determine whether canonical or non-canonical WNT signaling is activated, thereby affecting the cross talk between TACs and MSCs that controls mesenchymal tissue homeostasis. These two signaling pathways work in concert to keep different cell types in proper balance and help MSCs and TACs coordinate the homeostatic demands of the mouse incisor.

In conclusion, we have identified reciprocal interaction between MSCs and TACs using the adult mouse incisor as a model. Our findings illustrate how MSCs feedforward to TACs, and how in turn TACs produce crucial signals that sustain MSCs. Collectively, our results demonstrate that the maintenance of adult tissue homeostasis requires not only stem cells and progenitor

cells such as TACs, but also reciprocal cell–cell interactions between them. It is reasonable to view the stem cell niche as a local ecosystem that helps to maintain proper stem cell activity by promoting cross talk between stem cells and their progeny, which together preserve tissue homeostasis.

# Materials and methods

## Key resources table

| Reagent type (species) or resource | Designation | Source or reference | Identifiers | Additional information |
|---|---|---|---|---|
| Strain, strain background (*M. musculus*) | *Axin2-CreER^{T2}* | Jackson Laboratory | Stock No. 018867 RRID:IMSR_JAX: 018867 | |
| Strain, strain background (*M. musculus*) | *Dmp1-Cre* | Jackson Laboratory | Stock No. 023047 RRID:IMSR_JAX: 023047 | |
| Strain, strain background (*M. musculus*) | *Gli1-CreER^{T2}* | Jackson Laboratory | Stock No. 007913 RRID:IMSR_JAX: 007913 | |
| Strain, strain background (*M. musculus*) | *Gli1-LacZ* | Jackson Laboratory | Stock No. 008211 RRID:IMSR_JAX: 008211 | |
| Strain, strain background (*M. musculus*) | *Igf1r^{fl/fl}* | Jackson Laboratory | Stock No. 012251 RRID:IMSR_JAX: 012251 | |
| Strain, strain background (*M. musculus*) | *Ror2^{flox/flox}* | Jackson Laboratory | Stock No. 018354 RRID:IMSR_JAX: 018354 | |
| Strain, strain background (*M. musculus*) | *Rosa26^{<fs-tdTomato>}* | Jackson Laboratory | Stock No. 007905 RRID:IMSR_JAX: 007905 | |
| Strain, strain background (*M. musculus*) | *Wnt5a^{flox/flox}* | Jackson Laboratory | Stock No. 026626 RRID:IMSR_JAX: 026626 | |
| Strain, strain background (*M. musculus*) | *Wls^{flox/flox}* | Jackson Laboratory | Stock No. 012888 RRID:IMSR_JAX: 012888 | |
| Genetic reagent (*M. musculus*) | Anti-Axin2 probe | Advanced Cell Diagnostics | Cat# 400331 | |
| Genetic reagent (*M. musculus*) | Anti-Fgf10 probe | Advanced Cell Diagnostics | Cat#446371 | |
| Genetic reagent (*M. musculus*) | Anti-Fzd2 probe | Advanced Cell Diagnostics | Cat#404881 | |
| Genetic reagent (*M. musculus*) | Anti-Fzd4 probe | Advanced Cell Diagnostics | Cat#404901 | |
| Genetic reagent (*M. musculus*) | Anti-Fzd6 probe | Advanced Cell Diagnostics | Cat#404921 | |
| Genetic reagent (*M. musculus*) | Anti-Igf1r probe | Advanced Cell Diagnostics | Cat#417561 | |

*Continued on next page*

*Continued*

| Reagent type (species) or resource | Designation | Source or reference | Identifiers | Additional information |
|---|---|---|---|---|
| Genetic reagent (*M. musculus*) | Anti-Igf2 probe | Advanced Cell Diagnostics | Cat#437671 | |
| Genetic reagent (*M. musculus*) | Anti-Igfbp3 probe | Advanced Cell Diagnostics | Cat#405941 | |
| Genetic reagent (*M. musculus*) | Anti-Lrp5 probe | Advanced Cell Diagnostics | Cat#315791 | |
| Genetic reagent (*M. musculus*) | Anti-Ror2 probe | Advanced Cell Diagnostics | Cat#430041 | |
| Genetic reagent (*M. musculus*) | Anti-Wnt1 probe | Advanced Cell Diagnostics | Cat#401091 | |
| Genetic reagent (*M. musculus*) | Anti-Wnt2 probe | Advanced Cell Diagnostics | Cat#313601 | |
| Genetic reagent (*M. musculus*) | Anti-Wnt2b probe | Advanced Cell Diagnostics | Cat#405031 | |
| Genetic reagent (*M. musculus*) | Anti-Wnt3 probe | Advanced Cell Diagnostics | Cat#312241 | |
| Genetic reagent (*M. musculus*) | Anti-Wnt3a probe | Advanced Cell Diagnostics | Cat#405041 | |
| Genetic reagent (*M. musculus*) | Anti-Wnt4 probe | Advanced Cell Diagnostics | Cat#401101 | |
| Genetic reagent (*M. musculus*) | Anti-Wnt5a probe | Advanced Cell Diagnostics | Cat#316791 | |
| Genetic reagent (*M. musculus*) | Anti-Wnt5b probe | Advanced Cell Diagnostics | Cat#405051 | |
| Genetic reagent (*M. musculus*) | Anti-Wnt6 probe | Advanced Cell Diagnostics | Cat#401111 | |
| Genetic reagent (*M. musculus*) | Anti-Wnt7a probe | Advanced Cell Diagnostics | Cat#401121 | |
| Genetic reagent (*M. musculus*) | Anti-Wnt7b probe | Advanced Cell Diagnostics | Cat#401131 | |
| Genetic reagent (*M. musculus*) | Anti-Wnt8a probe | Advanced Cell Diagnostics | Cat#405061 | |
| Genetic reagent (*M. musculus*) | Anti-Wnt8b probe | Advanced Cell Diagnostics | Cat#405071 | |
| Genetic reagent (*M. musculus*) | Anti-Wnt9a probe | Advanced Cell Diagnostics | Cat#405081 | |
| Genetic reagent (*M. musculus*) | Anti-Wnt9b probe | Advanced Cell Diagnostics | Cat#405091 | |
| Genetic reagent (*M. musculus*) | Anti-Wnt10a probe | Advanced Cell Diagnostics | Cat#401061 | |
| Genetic reagent (*M. musculus*) | Anti-Wnt10b probe | Advanced Cell Diagnostics | Cat#401071 | |
| Genetic reagent (*M. musculus*) | Anti-Wnt11 probe | Advanced Cell Diagnostics | Cat#405021 | |
| Genetic reagent (*M. musculus*) | Anti-Wnt16 probe | Advanced Cell Diagnostics | Cat#401081 | |
| Antibody | Anti-β-actin (Rabbit monoclonal) | Cell Signaling Technology | Cat#4970S RRID:AB_2223172 | (1:2000) |
| Antibody | Anti-Amelx (Rabbit polyclonal) | Abcam | Cat# ab153915 | (1:100) |

*Continued on next page*

*Continued*

| Reagent type (species) or resource | Designation | Source or reference | Identifiers | Additional information |
|---|---|---|---|---|
| Antibody | Anti-β-catenin (Rabbit monoclonal) | Cell Signaling Technology | Cat#8814S RRID:AB_11127203 | IF (1:100), WB (1:2000) |
| Antibody | Anti-β-gal (Chicken polyclonal) | Abcam | Cat#ab9361 RRID:AB_307210 | (1:100) |
| Antibody | Anti-c-Myc (Rabbit monoclonal) | Abcam | Cat#ab32072 RRID:AB_731658 | (1:100) |
| Antibody | Anti-Ki67 (Rabbit monoclonal) | Abcam | Cat# ab16667; RRID:AB_302459 | (1:200) |
| Antibody | Anti-Wls (Chicken polyclonal) | Abcam | Cat#ab72385 RRID:AB_1269023 | (1:200) |
| Antibody | Anti-Chicken (Goat polyclonal) | Life Technologies | Cat#A-11039 RRID:AB_142924 | (1:200) |
| Antibody | Anti-Rabbit (Goat polyclonal) | Life Technologies | Cat#A-11011 RRID:AB_143157 | (1:200) |
| Antibody | Anti-Chicken (Goat polyclonal) | Life Technologies | Cat#A-11041 RRID:AB_2534098 | (1:200) |
| Commercial assay or kit | RNeasy Micro Kit | QIAGEN | Cat# 74004 | |
| Commercial assay or kit | Click-iT EdU Cell Proliferation Kit | Thermo Fisher Scientific | Cat# C10337 | |
| Software, algorithm | ImageJ | NIH | RRID:SCR_003070 | |
| Software, algorithm | GraphPad Prism | GraphPad Software | RRID:SCR_002798 | |

## Animals and procedures

*Axin2-CreER$^{T2}$*, *Dmp1-Cre*, *Gli1–CreER$^{T2}$*, *Gli1–LacZ*, *Igf1r$^{flox/flox}$*, *Ror2$^{flox/flox}$*, *Rosa26$^{<fs-tdTomato>}$*, *Wnt5a$^{flox/flox}$*, and *Wls$^{flox/flox}$* mouse strains were cross-bred as needed for this study. All mouse experiments were conducted in accordance with protocol 20299 approved by the Department of Animal Resources and the Institutional Animal Care and Use Committee of the University of Southern California.

Mice were housed in pathogen-free conditions, identified via ear tags, and analyzed in a mixed background. Tail biopsies were lysed through incubation at 55°C overnight in DirectPCR tail solution (Viagen 102 T) followed by 30 min of heat inactivation at 85°C prior to PCR-based genotyping (GoTaq Green MasterMix, Promega, and C1000 Touch Cycler, Biorad). Mice were euthanized by carbon dioxide overdose followed by cervical dislocation. All mice were used for analysis regardless of sex.

For CreER$^{T2}$ activation, tamoxifen (Sigma) was dissolved in corn oil (20 mg/ml) and injected intraperitoneally.

## Immunofluorescence and ISH

Mouse mandibles were dissected, fixed in 4% PFA overnight, and decalcified with 10% EDTA for 4 weeks. The tissues were next incubated with 15% sucrose for 2 hr and 30% sucrose overnight, and then embedded in OCT. Frozen tissue blocks were sectioned at 10 mm on a cryostat (Leica) and mounted on SuperFrost Plus slides (Fisher). The tissue sections were blocked for 1 hr at room temperature in blocking solution (Vector Laboratories). Sections were then incubated with primary antibodies diluted in blocking solution at 4°C overnight. After three washes with PBS, sections were incubated with secondary antibodies in blocking solution at room temperature for 1 hr. DAPI was used to stain nuclei and all images were acquired using a Keyence microscope (Carl Zeiss). Non-

immune immunoglobulins of the same isotype as the primary antibodies were used as negative controls.

ISH was performed using an RNAscope Multiplex Fluorescent Assay (Advanced Cell Diagnostics). Briefly, tissues were fixed in 4% PFA overnight at room temperature before cryosectioning into 10 µm sections, after which ISH was performed according to the manufacturer's instructions.

## MicroCT analysis

MicroCT analysis was performed using a SCANCO µCT50 device at the University of Southern California Molecular Imaging Center. Images were acquired with the X-ray source at 70 kVp and 114 µA. Images were generated at a resolution of 10 µm. Three-dimensional reconstruction was achieved using AVIZO 7.1 (Visualization Sciences Group).

## Laser capture microdissection

We euthanized 1-month-old $Axin2$-$CreER^{T2}$;$Rosa26^{<fs-tdTomato>}$ and $Gli1$-$CreER^{T2}$;$Rosa26^{<fs-tdTomato>}$ mice (n = 3 per group) 1 day after the injection of tamoxifen. After quickly dissecting out the mandible and rinsing in PBS and OCT, the samples were transferred into a mold filled with pre-cooled OCT and frozen using liquid nitrogen. Next, the samples were sectioned to 10 µm thickness and mounted on Polyethylenenaphthalate (PEN)-membrane slides (Zeiss). After UV treatment for 30 min, we performed LCM of the Axin2+ and Gli1+ cells using the Zeiss PALM Laser Capture Microdissection System.

## RNA sequencing

Incisor samples from LCM of 1-month-old $Axin2$-$CreER^{T2}$;$Rosa26^{<fs-tdTomato>}$ and $Gli1$-$CreER^{T2}$;$Rosa26^{<fs-tdTomato>}$ mice were collected for RNA isolation using an RNeasy Micro Kit (QIAGEN). The quality of RNA samples was determined using an Agilent 2100 Bioanalyzer and only those with RNA integrity (RIN) numbers >7.0 were used for sequencing. cDNA library preparation and sequencing were performed at the Epigenome Center of the University of Southern California. Single-end reads with 75 cycles were performed on an Illumina Hiseq 4000 for three pairs of samples. Raw reads were trimmed, aligned with the mm10 genome using TopHat (version 2.0.8), and normalized using RPKM (Reads Per Kilobase Million). Differential expression was calculated by selecting transcripts with a significance level set to p<0.05.

## Immunoblotting

For immunoblotting, mouse incisor mesenchyme was lysed in lysis buffer (50 mM Tris-HCl pH 7.5, 150 mM NaCl, 2 mM EDTA, 0.1% NP-40, 10% glycerol, and protease inhibitor cocktail). Proteins were quantified using Bio-Rad protein assay (Bio-Rad Laboratories), and 20–80 µg of protein was separated by SDS-PAGE and transferred to 0.45 µm PVDF membrane. Membranes were blocked in TBST and 5% BSA (blocking solution) for 1 hr, followed by overnight incubation with active β-catenin antibody diluted at 1:2000 in blocking solution, and 1 hr incubation with HRP-conjugated secondary antibody diluted at 1:2000. Immunoreactive protein was detected using ECL (GE Healthcare) and film.

## Label-retaining cell analysis

Mice were given i.p. injections of EdU (150 mg/kg) daily for 2 weeks. Samples were collected 1 month after the last injection and processed for further analysis with the Click-iT EdU Cell Proliferation Kit based on the manufacturer's protocol.

## Statistical analysis

Prism 8 (GraphPad) was used for statistical analysis. All bar graphs display mean ± SD (standard deviation). Significance was assessed by independent two-tailed Student's t-tests or ANOVA. p<0.05 was considered statistically significant.

## ImageJ image analysis

ImageJ was used to determine the percentage of immunostained area. Images of the TAC and MSC regions were first converted to 8-bit binary and evaluated for positive immunofluorescence signal

using the "Analyze Particles" function. The derived area was then divided by the total area of TAC or MSC regions to calculate the percentage of positive immunostaining.

## Acknowledgements

We thank Julie Mayo and Bridget Samuels for critical reading of the manuscript. We acknowledge USC Libraries Bioinformatics Service for assisting with data analysis. The bioinformatics software and computing resources used in the analysis are funded by the USC Office of Research and the Norris Medical Library. This study was supported by grants from the National Institute of Dental and Craniofacial Research and National Institutes of Health (R01 DE025221 and R01 DE012711).

## Additional information

### Funding

| Funder | Grant reference number | Author |
| --- | --- | --- |
| National Institute of Dental and Craniofacial Research | R01 DE025221 | Yang Chai |
| National Institute of Dental and Craniofacial Research | R01 DE012711 | Yang Chai |

The funders had no role in study design, data collection and interpretation, or the decision to submit the work for publication.

### Author contributions

Junjun Jing, Conceptualization, Resources, Data curation, Formal analysis, Investigation, Methodology, Writing - original draft, Project administration, Writing - review and editing; Jifan Feng, Data curation, Validation, Methodology; Jingyuan Li, Yuan Yuan, Paul Sharpe, Data curation, Methodology; Hu Zhao, Data curation, Formal analysis; Thach-Vu Ho, Jinzhi He, Tingwei Guo, Jiahui Du, Data curation; Mark Urata, Data curation, Visualization, Writing - original draft; Yang Chai, Conceptualization, Resources, Data curation, Formal analysis, Supervision, Funding acquisition, Validation, Investigation, Visualization, Methodology, Writing - original draft, Project administration, Writing - review and editing

### Author ORCIDs

Junjun Jing (iD) https://orcid.org/0000-0001-5745-5207
Thach-Vu Ho (iD) http://orcid.org/0000-0001-6293-4739
Paul Sharpe (iD) http://orcid.org/0000-0003-2116-9561
Yang Chai (iD) https://orcid.org/0000-0003-2477-7247

### Ethics

Animal experimentation: All mouse experiments were conducted in accordance with protocols 20299 approved by the Department of Animal Resources and the Institutional Animal Care and Use Committee of the University of Southern California.

### Decision letter and Author response

Decision letter https://doi.org/10.7554/eLife.59459.sa1
Author response https://doi.org/10.7554/eLife.59459.sa2

## Additional files

### Supplementary files

- Transparent reporting form

## Data availability

Sequencing data have been deposited in GEO under accession codes GSE109876.

The following dataset was generated:

| Author(s) | Year | Dataset title | Dataset URL | Database and Identifier |
|---|---|---|---|---|
| Jing J, Feng J | 2019 | Bi-directional interaction between mesenchymal stem cells and transit amplifying cells in mesenchymal tissue homeostasis [MSCTAC] | https://www.ncbi.nlm.nih.gov/geo/query/acc.cgi?acc=GSE109876 | NCBI Gene Expression Omnibus, GSE109876 |

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
