## [Decision Letter]

**Acceptance summary:**

This work is a genetic tour de force and I appreciate the outstanding effort in the additional materials that support the genetic models you report. Significant new evidence regarding the genetic pathways regulating MSCs and TACs in the incisor constitutes a significant addition to current knowledge.

**Decision letter after peer review:**

Thank you for submitting your article "Reciprocal interaction between mesenchymal stem and transit amplifying cells in tissue homeostasis". Your article has been reviewed by three reviewers, including Deneen Wellik as the Reviewing Editor and Reviewer #1, and the evaluation has been overseen by a Senior Editor. The reviewers additionally had significant discussions regarding the review and Dr Wellik has drafted this decision to help you prepare a revised submission.

In this manuscript, the authors use a variety of genetic models to provide evidence for the MSC-TAC reciprocal signaling crosstalk. They propose that IGF-Wnt is involved in feedforward from MSCs to TACs, with tissue autonomous WNT/Ror2 signaling in TACs providing feedback to MSCs.

This manuscript was reviewed and significant new data that impacts the field is reported, however, there were concerns regarding the manuscript in its current state. Some of these concerns were placated during the reviewer consultation phase, but others remain, and we request that the authors address these concerns prior to publication. We would like to draw your attention to changes in our revision policy that we have made in response to COVID-19 (https://elifesciences.org/articles/57162). First, because many researchers have temporarily lost access to the labs, we will give authors as much time as they need to submit revised manuscripts. We are also offering, if you choose, to post the manuscript to bioRxiv (if it is not already there) along with this decision letter and a formal designation that the manuscript is “in revision at *eLife*”. Please let us know if you would like to pursue this option. (If your work is more suitable for medRxiv, you will need to post the preprint yourself, as the mechanisms for us to do so are still in development.)

Essential Revisions:

The overall strong concerns are regarding the interpretation of some of the results without additional controls or time points for the genetic models:

The conditional LOF experiments do not include rigorous validation of efficient deletion (*Igf1r*, *Wntless*, *Wnt5a*).

For genetic manipulations that affect only the TAC population, one would expect the resulting phenotype to be short-term – until the MSCs (in which the genetic manipulation did not occur) replace the TAC population. These experiments should be included to support their interesting conclusions. This is particularly important given that all three reviewers noted that the genetic manipulations do not appear to be “clean”, i.e. expression and perturbations do not appear perfectly aligned to the MSC or TAC populations, but are visualized in the epithelium, for instance, and other surrounding regions, making it difficult to conclude with confidence cause for the phenotype without more rigorous examination.

Related, the basis of IGF LOF accelerated and increased differentiation and decreased proliferation, and longer-term experiments to show the resolution expected if the population proposed is affected.

Additionally, if Wnts play a trophic role on TACs, this should be supported by demonstrating survival changes in this population at various time points.

With regards to quantifying MSC populations, more than one marker (Gli1) should be included. The identity and changes in this cell population is key to their model and additional markers should be assessed.

A Gli1 null to control for the genetic model of Gli1CreER/LacZ is needed. The authors report a Gli1LacZ het as the control, but this does not control for LOF at this allele that is created by use of Cre and LacZ at this locus.

These comments speak to an overall concern that the authors need to more carefully rule in or out epithelial-mesenchymal crosstalk as opposed to MSC-TAC crosstalk that the authors assert. Many of the expression patterns appear to include epithelium and whether LOF was enacted in this population (or this population is impacted) is not clear. More carefully reporting the extent of deletions and including later time points that behave as expected if functional deletions are affecting the relevant populations would strengthen this work.

---

## [Author Response]

Essential Revisions:The overall strong concerns are regarding the interpretation of some of the results without additional controls or time points for the genetic models:The conditional LOF experiments do not include rigorous validation of efficient deletion (Igf1r, Wntless, Wnt5a).

We thank the reviewer for this suggestion. We have confirmed the efficient deletion of *Igf1r,Wnt5a* and *Wntless* in *Axin2-CreER^T2^;Igf1r^fl/fl^*, *Axin2-CreER^T2^;Wnt5a^fl/fl^* and *Axin2-CreER^T2^;Wntless^fl/fl^* mutant models (Figure 5—figure supplement 1 in the manuscript and Author response image 1). The results indicate that these genes are efficiently deleted in the TAC region in the mesenchyme of these mutants.

**Author response image 1. sa2fig1:** RNAscope staining of *Igf1r* (A-D) and *Wntless* (E-H) in control, *Axin2-CreER^T2^;Igf1r^fl/fl^* and *Axin2-CreER^T2^;Wntless^fl/fl^* mutant incisors three days after tamoxifen induction. Arrow indicates positive signal and asterisk indicates absence of signal.

For genetic manipulations that affect only the TAC population, one would expect the resulting phenotype to be short-term – until the MSCs (in which the genetic manipulation did not occur) replace the TAC population. These experiments should be included to support their interesting conclusions.

We appreciate this comment. Based on our study, TACs are a critical niche component of MSCs in the mouse incisor. Thus, even though the TAC defects are short-term, the subsequent effect on MSCs’ niche environment will impact MSC fate and lead to a long-lasting disturbance to the homeostasis of the incisor. Indeed, the phenotype is long-lasting in inducible knockouts (*Axin2-CreER^T2^;Igf1r^fl/fl^*, *Axin2CreER^T2^;Wnt5a^fl/fl^* and *Axin2-CreER^T2^;Wntless^fl/fl^*) in which mutations only occur in Axin2+ TACs during short-term tamoxifen induction. More specifically, within a short period (one week after tamoxifen induction), the number of TACs in the incisor mesenchyme was already reduced in *Axin2CreER^T2^;Igf1r^fl/fl^;Gli1-LacZ* mice (Figure 3—figure supplement 1 in the manuscript) whereas the number of Gli1+ MSCs was unchanged in *Axin2-CreER^T2^;Igf1r^fl/fl^;Gli1-LacZ* incisors compared to those of *Gli1-LacZ* incisors at this time point (Author response image 2). However, after a longer period (three weeks after tamoxifen induction), the number of Gli1+ MSCs was reduced, suggesting the change of TACs in the mesenchyme has an adverse effect on MSCs at a later stage through TAC-MSC feedback in this mutant model (see Author response image 2). Because the MSCs were impacted and tissue homeostasis was disturbed in *Axin2-CreER^T2^;Igf1r^fl/fl^* incisors, the phenotype could persist for five months after tamoxifen induction in this mutant model (Figure 3—figure supplement 3 in the manuscript).

Consistent with the above findings, the number of MSCs was also reduced while only the TAC-derived Wnts or *Wnt5a* were affected in *Axin2-CreER^T2^;Wntless^fl/fl^;Gli1-LacZ* (Figure 4—figure supplement 3 in the manuscript) and *Axin2-CreER^T2^;Wnt5a^fl/fl^;Gli1-LacZ* mice three weeks after tamoxifen induction (Figure 5A-C in the manuscript). We have also further confirmed that the phenotype in *Axin2-CreER^T2^;Wnt5a^fl/fl^* (Figure 5—figure supplement 3 in the manuscript) and *Axin2-CreER^T2^;Wntless^fl/fl^* (Author response image 3) incisors persisted for five months after tamoxifen induction.

**Author response image 2. sa2fig2:** (A-D) Immunostaining of β-gal in *Gli1-lacZ* and *Axin2-CreER^T2^;Igf1r^fl/fl^;Gli1-LacZ* incisors one week after tamoxifen induction. (E) Quantitative analysis of Gli1+ MSCs of *Gli1-lacZ* and *Axin2-CreER^T2^;Igf1r^fl/fl^;Gli1-LacZ* incisors one week after tamoxifen induction. (F-I) Immunostaining of β-gal in *Gli1-lacZ* and *Axin2CreER^T2^;Igf1r^fl/fl^;Gli1-LacZ* incisors three weeks after tamoxifen induction. (J) Quantitative analysis of Gli1+ MSCs of *Gli1-lacZ* and *Axin2-CreER^T2^;Igf1r^fl/fl^;Gli1-LacZ* incisors three weeks after tamoxifen induction.

**Author response image 3. sa2fig3:** (A-D) H & E staining and Ki67 immunostaining of *Wntless^fl/fl^* and *Axin2-CreER^T2^;Wntless^fl/fl^* incisors five months after tamoxifen induction. Black arrow in (A) indicates normal dentin and black arrow in (C) indicates thicker dentin.

This is particularly important given that all three reviewers noted that the genetic manipulations do not appear to be “clean”, i.e. expression and perturbations do not appear perfectly aligned to the MSC or TAC populations, but are visualized in the epithelium, for instance, and other surrounding regions, making it difficult to conclude with confidence cause for the phenotype without more rigorous examination.

We thank the reviewer for this comment. In order to examine whether the incisor epithelium has an effect on the observed phenotype, we further analyzed the change in the incisor epithelium overtime. Specifically, one week after tamoxifen induction, the number and differentiation status of TACs in the incisor epithelium remained unchanged in *Axin2-CreER^T2^;Igf1r^fl/fl^* mice at this time point, while the number of TACs in the incisor mesenchyme was already reduced and TAC differentiation was enhanced, suggesting that the effect from the dental mesenchyme is primary (Figure 3—figure supplement 1 in the manuscript). Later on, three weeks after tamoxifen induction, the number of TACs in the incisor epithelium began to reduce in *Axin2CreER^T2^;Igf1r^fl/fl^*mice (Figure 3C-D in the manuscript), suggesting that epithelial-mesenchymal interaction occurs later and that the epithelial defect is secondary. Similarly, TACs in the mesenchyme of *Axin2-CreER^T2^;Wntless^fl/fl^* incisors were reduced one week after tamoxifen induction but the number of TACs in the epithelium of the mutant incisors was not changed until three weeks after tamoxifen induction (Author response image 4 and Figure 4—figure supplement 3 in the manuscript). Likewise, the number of TACs in the epithelium began to reduce after the change of MSCs was observed in *Axin2-CreER^T2^;Wnt5a^fl/fl^;Gli1LacZ* incisors (Author response image 5 and Figure 5—figure supplement 3 in the manuscript).

**Author response image 4. sa2fig4:** (A-D) H & E staining and Ki67 immunostaining of *Wntless^fl/fl^* and *Axin2-CreER^T2^;Wntless^fl/fl^* incisors. (E-F) Quantitative analysis of Ki67+ TACs in the mesenchyme and epithelium of *Wntless^fl/fl^* and *Axin2-CreER^T2^;Wntless^fl/fl^* incisors one week after tamoxifen induction.

**Author response image 5. sa2fig5:** (A-B) Ki67 immunostaining of *Gli1-LacZ* and *Axin2-CreER^T2^;Wnt5a^fl/fl^;Gli1-LacZ* incisors three weeks after tamoxifen induction. (C-D) Quantitative analysis of Ki67+ TACs in the mesenchyme and epithelium of *Gli1-LacZ* and *Axin2-CreER^T2^;Wnt5a^fl/fl^;Gli1-LacZ* incisors three weeks after tamoxifen induction.

Related, the basis of IGF LOF accelerated and increased differentiation and decreased proliferation, and longer-term experiments to show the resolution expected if the population proposed is affected.

We thank the reviewer for this suggestion. We have collected the samples from *Igf1r^fl/fl^* and *Axin2CreER^T2^;Igf1r^fl/fl^* mice five months after tamoxifen induction. We found that TACs were reduced in the incisor mesenchyme and more dentin formed in *Axin2-CreER^T2^;Igf1r^fl/fl^*mice, suggesting premature differentiation of TACs five months after tamoxifen induction (Figure 3—figure supplement 3 in the manuscript).

Additionally, if Wnts play a trophic role on TACs, this should be supported by demonstrating survival changes in this population at various time points.

We thank the reviewer for this suggestion. We have examined cell apoptosis by TUNEL assay at two different time points in *Axin2-CreER^T2^;Wntless^fl/fl^* mice (Author response image 6). The results suggest that loss of WNT signaling in TACs leads to increased cell apoptosis in the TAC region of the incisor mesenchyme.

**Author response image 6. sa2fig6:** (A-H) TUNEL staining of *Wntless^fl/fl^* and *Axin2-CreER^T2^;Wntless^fl/fl^* incisors one week and 3 weeks after tamoxifen induction. Arrow indicates positive signal and asterisk indicates absence of signal.

With regards to quantifying MSC populations, more than one marker (Gli1) should be included. The identity and changes in this cell population is key to their model and additional markers should be assessed.A Gli1 null to control for the genetic model of Gli1CreER/LacZ is needed. The authors report a Gli1LacZ het as the control, but this does not control for LOF at this allele that is created by use of Cre and LacZ at this locus.

We appreciate this suggestion. We have used label retaining cells (LRCs), a hallmark of stem cells, to quantify MSC populations in this study, and we have added the data into Figure 5—figure supplement 2 and Figure 6—figure supplement 1 in this manuscript. We found that the results were consistent with those of the Gli1+ MSCs. We have used *Gli1-CreER^T2^;Gli1-LacZ* mice as the control to compare the MSCs with *Gli1CreER^T^;Ror2^fl/fl^;Gli1-LacZ* mice (Author response image 7). Our data suggest that loss of Gli1 has no impact on the number of Gli1+ MSCs.

**Author response image 7. sa2fig7:** (A-F) Immunostaining of β-gal in *Gli1-LacZ*, *Gli1-CreER^T2^;Gli1-LacZ* and *Gli1-CreER^T2^;Ror2^fl/fl^;Gli1LacZ* incisors 3 weeks after tamoxifen induction. (G) Quantitative analysis of Gli1+ MSCs in the mesenchyme of *Gli1-LacZ*, *Gli1-CreER^T2^;Gli1-LacZ* and *Gli1-CreER^T2^;Ror2^fl/fl^;Gli1-LacZ* incisors at higher magnification (B, D and F) 3 weeks after tamoxifen induction.

These comments speak to an overall concern that the authors need to more carefully rule in or out epithelial-mesenchymal crosstalk as opposed to MSC-TAC crosstalk that the authors assert. Many of the expression patterns appear to include epithelium and whether LOF was enacted in this population (or this population is impacted) is not clear. More carefully reporting the extent of deletions and including later time points that behave as expected if functional deletions are affecting the relevant populations would strengthen this work.

We thank the reviewer for these suggestions. *Axin2-CreER^T2^* is highly active in the TACs of the incisor mesenchyme but not epithelium, and our data have shown that the genes were efficiently deleted in Axin2+ TACs of the incisor mesenchyme but not epithelium. As we discussed in our responses to earlier questions, we have shown that epithelial-mesenchymal crosstalk in the incisor indeed occurs only later and is secondary, whereas MSC-TAC interaction occurs first and is the primary effect underlying the phenotype observed in this study. Because the homeostasis of the incisor is disturbed due to the interrupted MSC-TAC crosstalk, the phenotype is long-lasting.